# Does County-Level Medical Centre Policy Influence the Health Outcomes of Patients with Trauma Transported by the Emergency Medical Service System? An Integrated Emergency Model in Rural China

**DOI:** 10.3390/ijerph16010133

**Published:** 2019-01-06

**Authors:** Dai Su, Yingchun Chen, Hongxia Gao, Haomiao Li, Jingjing Chang, Shihan Lei, Di Jiang, Xiaomei Hu, Min Tan, Zhifang Chen

**Affiliations:** 1School of Medicine and Health Management, Tongji Medical College, Huazhong University of Science and Technology, Wuhan 430030, China; sudai@hust.edu.cn (D.S.); gaohongxia@hust.edu.cn (H.G.); lihaomiao@hust.edu.cn (H.L.); changjingjing@hust.edu.cn (J.C.); leishihan@hust.edu.cn (S.L.); jiangdi@hust.edu.cn (D.J.); huxiaomei@hust.edu.cn (X.H.); tanmin@hust.edu.cn (M.T.); m201875289@hust.edu.cn (Z.C.); 2Research center for Rural Health Services, Hubei Province Key Research Institute of Humanities and Social Sciences, Wuhan 430030, China

**Keywords:** county-level medical centre, health outcomes, patients with trauma transported by EMS, propensity score matching, Cox proportional hazard model

## Abstract

This study aimed to assess the effect of the county-level medical centre policy on the health outcomes of trauma patients transported by emergency medical service (EMS) system in rural China. The methodology involved the use of electronic health records (EHRs, after 2016) of patients with trauma conditions such as head injury (*n* = 1931), chest (back) injury (*n* = 466), abdominal (waist) injury (*n* = 536), and limb injury (*n* = 857) who were transported by EMS to the county-level trauma centres of Huining County and Huan County in Gansu, China. Each patient was matched with a counterpart to a county-level trauma centre hospital by propensity score matching. Cox proportional hazard models were used to estimate the hazard ratios (HRs) of such patients in different hospitals. The HRs of all patients with the abovementioned traumatic conditions transported by EMS to county-level trauma centre hospitals were consistently higher than those transported by EMS to traditional hospitals after adjusting for numerous potential confounders. Higher HRs were associated with all patients with trauma (HR = 1.249, *p* < 0.001), head injury (HR = 1.416, *p* < 0.001), chest (back) injury (HR = 1.112, *p* = 0.560), abdominal (waist) injury (HR = 1.273, *p* = 0.016), and limb injury (HR = 1.078, *p* = 0.561) transported by EMS to the county-level trauma centre hospitals. Our study suggests that the construction of county-level medical centre provides an effective strategy to improve the health outcomes of EMS-transported trauma patients in Gansu, China. Policy makers can learn from the experience and improve the health outcomes of such patients through a personalised trauma treatment system and by categorizing the regional trauma centre.

## 1. Introduction

Traumatic injury and trauma deaths have become a major global health problem, and they are the most frequent cause of death amongst people under 45 years old [1]. The World Health Organization estimates that 16,000 people die every day from trauma injuries; for every person who dies, several thousands more are injured, and many of them have permanent sequelae. Injury accounts for 16% of the global burden of disease [2]. As the most populous country worldwide with an accelerated development of urbanization and transportation, China has recorded traumatic death as the fifth cause of human deaths. Compared with developed countries such as the United States and the United Kingdom, the rate of traumatic death is twice as high in China [3]. Patients with trauma transported by EMS have reached an alarming proportion, representing approximately 35%–40% in all patients transported by EMS [4]. Head injury, chest (back) injury, abdominal (waist) injury, and limb injury are the most common trauma types of patients transported by EMS in China.

Evidence from the United Kingdom, Germany, France, and Australia suggests that trauma centre-based treatment systems can reduce the overall mortality of patients with trauma by 8–15% [5,6,7]. In the United States during 1960s, the earliest trauma centre was established. In the 1980s, the American College of Surgeons (ACS) developed a classification system for trauma centres, and different levels of trauma centres (divided into 1–3, with level 1 being the highest) were established according to hospitals with different capabilities. Referrals and technical guidance were established between the senior- and low-level trauma centres. Each centre has an integrated business department that is independent from other centres. This approach has significantly improved the survival rate of EMS patients with trauma [8]. Since the establishment of a comprehensive trauma centre in 1969, the number of traffic accident deaths in Germany has dropped from more than 20,000 in 1970 to 4000 in 2011 [9]. In view of the substantial increase in the incidence and mortality rates of trauma in China, hospitals must integrate multidisciplinary collaborations such as pre-hospital, emergency, orthopaedics, neurosurgery, and critical care, to provide comprehensive treatments. Therefore, trauma centres were constructed in China in the 1990s. However, they were mainly built in urban hospitals, especially large-scale hospitals, which integrate the resources needed for trauma treatment, practice a comprehensive treatment approach, improve the efficiency of treatment, enhance the prognosis, and reduce the death rate and disability rate of patients with trauma [10]. However, the construction of trauma centres in rural areas in China is lagging behind that in urban areas, and these areas need trauma centres. The mortality of injury and poisoning of rural residents in 2016 was 54.48/100,000, which is considerably higher than the city rate of 37.34/100,000. Many potential delay factors in rural areas lead to high morbidity and mortality. The trauma emergency delivery system in counties faces many challenges [11]. Firstly, many trauma patients are located in areas that are difficult to reach, thereby requiring long-term search and rescue. Secondly, rural mobile communication networks may not fully cover all areas, and patients and hospitals lack effective communication. Third, the medical conditions in township hospitals are backward; patients with traumatic emergency cannot acquire rapid treatment, and transit to the county hospital delays treatment. In addition, a hospital’s emergency area has structural shortcomings. Even with the most powerful county people’s hospital, medical trauma staff and patients are scattered in various clinical departments, and providing initial comprehensive treatment plans is difficult.

In response to the above problems in the rural EMS system and the trauma treatment system, in January 2018, the Chinese government issued the Notice on Printing and Distributing the Action Plan for Further Improving Medical Services (2018–2020), which clarifies that five medical centres with strong rescue capabilities will be installed in county-level hospitals and that diagnosis and treatment grading systems and mechanisms for cardiovascular diseases, severe trauma, critical maternal cases, and critically ill new-borns will be established. Such establishment will be led by tertiary hospitals and supported by primary hospitals and information technology. This approach will optimise the collaborative service process between pre-hospital emergency and in-hospital emergency departments and other departments, and provide a green channel of medical treatment and integrated comprehensive treatment services. As one of the five medical centres, the trauma centre should be constructed quickly, in addition to a joint emergency centre, to establish a county trauma treatment network that can effectively improve the ability to treat traumatic diseases, improve the success rate of treatment, and reduce mortality.

The evaluation and management of patients with trauma transported by EMS [12,13,14] and the impact of EMS and non-EMS transport on the survival rate of patients with trauma [15,16,17] have been extensively studied, but the impact of changes in the trauma treatment system on the treatment outcome of patients with trauma transported by EMS remains unknown. In addition, previous studies on the impact of trauma centres on treatment outcomes were limited to the urban areas in China [18,19]. Individual variability in different patients may be overlooked due to deficiencies in statistical methods, thereby possibly resulting in estimation bias and providing false evidence in the process of policy development and evaluation.

This study aimed to investigate the impact of constructing medical centres in county-level hospitals in China on the health outcomes of patients with trauma transported by EMS. In January 2016, Gansu Provincial Health and Family Planning Commission launched the pilot construction of five medical centres in county-level hospitals. Two county-level people’s hospitals in Huining (pilot county) and Huan (non-pilot county) with similar conditions were selected as the sample. Based on the standardised EHRs of patients with head injury, chest (back) injury, abdominal (waist) injury, and limb injury, we compared the cumulative cure and improvement probability and the relative risk of patients with trauma transported by EMS in the pilot county with those in the non-pilot county.

## 2. Materials and Methods

### 2.1. Study Design and Sampling

Gansu Province is located in Northwestern China. We chose Huining County and Huan County, which are located in the eastern part of Gansu Province (Figure 1). In each county, the largest county-level hospital was chosen as our sample hospital. The two hospitals were chosen for two reasons, as follows. (1) Firstly, Huining People’s Hospital is a pilot hospital for the construction of medical centres in Gansu. In January 2016, a trauma centre was established in Huining People’s Hospital, whereas the Huan People’s Hospital has not yet established a trauma centre; (2) Secondly, the two counties are similar in terms of the economic development level, the population of the area, and the medical treatment capacity in the two sample hospitals (Table 1).

This study is based on the standardised EHRs in the Gansu Provincial Health and Family Planning Commission database. We collected data on patients with trauma who were transported by EMS at Huining County People’s Hospital and Huan County People’s Hospital two and a half years after the county-level medical centre pilot study (January 2016 to June 2018). The International Classification of Diseases 10th Edition (ICD-10) is used to determine the diagnosis of head injury (S00.001-S09.906), chest injury (S20.201-S29.951), abdominal injury (S30.001-S39. 912), and limb injury (S41.001-S99.901). Personal identification numbers, such as name and ID number, of all patients were excluded before the study began. The data included demographic characteristics (e.g., age and gender), admission status, diagnostic code and the number of diagnoses (ICD-10 code for the patient’s primary diagnosis and up to 10 secondary diagnoses), inpatient care costs, surgery, and outcomes (information about the discharge result). After observations selection, a total of 1931 patients with head injury, 466 with chest (back) injury, 536 with abdominal (waist) injury, and 857 with limb injury were identified in the study (Figure 2).

### 2.2. Variables

#### 2.2.1. Outcome Variables

Considering the strong family relationships in rural China and the culture of filial piety and hospice care, if death is the outcome variable, rural patients may choose to die at home, and research may be biased. This problem is critical and should be considered when studying patients with trauma in rural China. More than 90% of patients with trauma transported by EMS in China can eventually be cured or improved to meet discharge standards [20,21]. Therefore, we used binary variable recovery as an outcome variable when patients were discharged. We recorded an EMS-transported patient with trauma who was cured or improved to meet the discharge criteria as 1 (positive result event), whereas those who died and those who did not meet the discharge criteria were recorded as 0 (negative result event).

#### 2.2.2. Explanatory Variables

We set the age group of 18–40 years old as the control group, because this age group has a high incidence of trauma, and because 20–40-year-old individuals have better physical health and resilience to in cases of emergency. Through the emergency transfer of ambulances, this age group has a better cure and improvement rate than other age groups. Other age groups were classified at 10-year intervals, and older EMS-transported patients with trauma generally had poorer health outcomes. The incidence and prognosis of traumatic first-aid patients with different genders are significantly different. Typically, the ratio of male and female trauma emergency patients is approximately 3:1; male patients generally are in a more serious condition, with a worse prognosis than females [22]. Admission status is another factor that affects the prognosis of such patients. Patients with trauma classified as “urgent” generally have worse health outcomes than regular patients. The complications and the risk of nosocomial infection affect the health outcomes, whether or not surgery is performed. The inpatient care cost represents the complexity of the condition of the EMS-transported patients with trauma; the higher the cost of inpatient care, the worse the health outcomes are. The number of disease diagnoses has an important impact on the prognosis of patients with traumatic emergency. Our main basis for screening patients with trauma is the main diagnosis of the first page of the EHRs. Oftentimes, the greater the number of diseases diagnosed, the higher the possibility of multiple trauma. The number of disease diagnoses has been used to predict the severity of disease in such patients. Therefore, we used the number of disease diagnoses as a measure of the severity of the disease. The number of disease diagnosis was indicated by a continuity score between 0 and 9, with larger values representing higher disease severity.

### 2.3. Statistical Analysis

All covariates were considered as categorical variables, except sex, admission status and surgery, which were treated as continuous variables. To estimate the impact of the construction of a county-level hospital medical centre on the health outcomes of EMS-transported patients with trauma, we compared the results between the treatment group and the comparable control group. Although we selected two similar hospitals in the hospital sample and could easily select treatment groups from the EHRs, some characteristics of the control group may not be consistent with those of the treatment group. To solve this shortcoming, we used propensity score matching (PSM) to build a more suitable control group [23]. The PSM method is widely used to estimate the impact of policy interventions on health outcomes, whilst randomised controlled trials are not feasible. Matching, which is based on propensity scores, can identify individuals in the control group with similar characteristics as those affected by the policy.

Before estimating the propensity score based on a rich set of covariates for logistic regression, we initially limited the potential control group by considering the key individual characteristics and data availability of the treatment group. Given that the number of EMS-transported patients with trauma between January 2016 and June 2018 in the two hospitals was similar, k-nearest neighbour matching (1:1) was used for our main analysis [24]. Observations with no common support were excluded from the analysis. We checked the balance of means of covariates after matching by examining the standardised mean differences between the control group and the treatment group (i.e., the baseline is also checked before and after matching). After matching, the bias should be ≤5% (or *p*-value >0.1) to establish adequate matching. Then, we calculated the achieved percentage of the bias reduction and examined each covariate’s standardised bias percentage before and after matching. Finally, we determined whether the k-nearest neighbour matching (1:1) is appropriate based on the mean reduction bias and median reduction bias of overall balance.

In general, the distribution of length of stay (LoS) is generally skewed, and the average LoS can be misleading; hence, we use the kernel density plot to display the LoS distribution and further analyse the LoS distributions in the various groups of trauma patients. LoS is an excellent proxy for severity and patient resilience not otherwise encapsulated in clinical codes. However, if appropriate care is delayed or not given it can also result in increased LoS.

Kaplan–Meier curves were firstly plotted to show the trend of the cumulative events of patients after matching. Cox proportional hazard regression models were used to calculate the HRs for different types of patients experiencing trauma for age, gender, admission status, inpatient care cost, surgery and the number of disease diagnoses. To further assess the validity of the Cox proportional hazard model, we tested the proportional hazard assumption whether a significant correlation existed between the rank of Schoenfeld residuals and the survival time for each covariate [25,26]. If the covariate did not satisfy the professional hazard hypothesis, we added the covariate–time interaction to the model. Two-sided *p*-values lower than 0.05 were considered statistically significant. We used R software version 3.5.1 (package survival and survminer) for data visualisations, and STATA (v. 14.0, StataCorp, College Station, TX, USA) for all data analyses.

### 2.4. Ethical Statement

This study was approved by the Ethics Committee of the Tongji Medical College, Huazhong University of Science and Technology (IORG No: IORG0003571).

## 3. Results

### 3.1. Patient Characteristics

We first examined the balancing property of each observed covariate between the treatment and control groups and the reduced sampling bias achieved through k-nearest neighbour matching (1:1). The results suggest that almost all observable covariates are sufficiently balanced by the matching between the treatment and control groups. The initial differences in the two groups are reduced considerably and have become statistically insignificant at 5%.

Table 2 shows the characteristics of EMS-transported patients with traumatic conditions such as head injury, chest (back) injury, abdominal (waist) injury, and limb injury, in the treatment group and the control group before and after matching. A total of 2132 patients with trauma comprised the control group before matching, higher than the 1658 patients with trauma in the treatment group. The number of patients with four individual trauma types in the control group exceeded the number of those with trauma emergency treatment in the treatment group. At the same time, a large difference was observed in the characteristics of the head injury, chest (back) injury, abdominal (waist) injury and limb injury between the treatment group and the control group before matching. The age structure of patients with chest (back) injury in the treatment group and the control group differed. The control group had a lower number of disease diagnoses, and the percentage of patients in the normal state during admission was higher than in the treatment group, thereby indicating that a larger proportion of patients with severe trauma were transferred to Huining County People’s Hospital. Although the differences in the characteristics of the four trauma types were significantly reduced after matching, the characteristics of each covariate still differed.

### 3.2. Distribution of LoS between Different Kinds of Trauma Patients in Two Groups 

Figure 3 shows the kernel density curves of LoS for matched head injury, chest (back) injury, abdominal (waist), and limb injury patients transported by EMS in two groups. We found that the peaks of curves of four types of trauma patients transported by EMS were more towering and shifted to the left to varying degrees, and the area under the tail of the curves was reduced, indicating a reduction in LoS after policy implementation.

### 3.3. Kaplan–Meier Survival Curves of Patients with Matched Data

Figure 4 shows the Kaplan–Meier survival curves for full-sample patients with matched data, with the yellow line indicating the patients with trauma in the treatment group and the blue line indicating the patients with trauma in the control group. The figure shows that compared with that of the control group at the same survival time, the survival probability of the treatment group is higher overall (chi-squared = 18.7, *p* < 0.001).

Similarly, Figure 5 shows the Kaplan–Meier survival curves and matching data for patients with head, chest (back), abdominal (waist), and limb injuries. Then, we compared the survival rates of the treatment group and the control group by log-rank test. The significant differences were noted in the survival rates of patients with head injury (chi-squared = 13.4, *p* < 0.001) and abdominal (waist) injury (chi-squared = 4.8, *p* = 0.030) in the two groups, whereas the difference in the survival rate of patients with chest (back) injury (chi-squared = 0.7, *p* = 0.400) and limb injury (chi-squared = 2.2, *p* = 0.100) in the two groups was not significant. Meanwhile, overall, those patients with trauma treated by medical centres had higher probability of cure and improvement than those in the control group with the same LoS.

### 3.4. Cox Hazard Model Estimates for the Four Trauma Types of Patients

Table 3 illustrates the estimates of the Cox hazard proportion model for all matched patients with head injury, chest (back) injury, abdominal (waist) injury, and limb injury. After using the Schoenfeld residuals to test the proportional hazard assumptions, four variables (admission status, inpatient care cost, surgery and the number of disease diagnoses) for all patients, two variables (inpatient care cost and the number of disease diagnoses) for those with head injury, one variable (inpatient care cost) for those with abdominal (waist) injury, and one variable (surgery) for those with limb injury did not meet the proportional hazard assumption. Therefore, interaction terms with LoS were added at each end of the columns of these variables. Higher HRs were associated with the treatment group across all patients with trauma (HR = 1.249, *p* < 0.001), head injury (HR = 1.416, *p* < 0.001), chest (back) injury (HR = 1.112, *p* = 0.560), abdominal (waist) injury (HR = 1.273, *p* = 0.016), and limb injury (HR = 1.078, *p* = 0.561).

## 4. Discussion

After the two groups of samples were processed by the PSM method, some features were controlled and met the proportion hazard assumption. The positive probability of all inpatients with trauma (HR = 1.249, 95% CI: 1.125–1.385, *p* < 0.001), head injury (HR = 1.416, 95% CI: 1.216–1.649, *p* < 0.001), chest (back) injury (HR = 1.112, 95% CI: 0.779–1.587, *p* = 0.560), abdominal (waist) injury (HR = 1.273, 95% CI: 1.021–1.848, *p* = 0.016), and limb injury (HR = 1.078, 95% CI: 0.838–1.387, *p* = 0.561) transported by EMS to the county-level hospital with medical centre construction (Huining County) was higher than those transported to county-level hospital (Huan County) with no trauma centres. In addition, the peaks of curves of four types of trauma patients transported by EMS were more towering and shifted to the left to varying degrees, and the area under the tail of the curves was reduced in Huining County.

Compared with the traditional trauma treatment system in Huan County, Huining County People’s Hospital, as the only hospital in the county to perform trauma centre pilots, integrates trauma-related department personnel and medical treatment projects within the hospital, implementing standardised treatment and optimising the diagnosis and treatment process. To ensure the labour ability and the quality of life of patients with trauma emergency, doctors need to be proficient in the key core technologies of various types of trauma treatment to provide diagnosis, assessment, treatment and transportation for patients with trauma emergency. The EMS is mainly responsible for the evaluation, initial treatment and transportation of patients with trauma in the pre-hospital stage. The trauma centre also maintains information communication with the EMS, provides high-efficiency technical guidance, and conducts rapid and comprehensive preliminary assessment after receiving the consultation. The trauma centre is also connected with the municipal blood station to establish a continuous 24-h blood supply mechanism to ensure the supply of blood products. Acute trauma is a complex disease that is often not limited to one site, may be multiple, or has other complications, usually requiring medical expertise and multidisciplinary coordination. Multidisciplinary joint consultations can be performed in a timely manner, but traditional in-hospital emergency doctors are weak in technology and provide inaccurate assessments, resulting in inefficient treatments. Given that doctors and accompanying staff are from different departments, obtaining their application forms for the trauma centre and the special identification of green channel can more easily connect with the relevant examination departments, thereby ensuring that trauma patients are rationally checked and transferred accordingly. Critically ill patients with trauma can be transported in a timely manner to the relevant departments for treatment, whereas the traditional in-hospital trauma treatment system needs communication for a longer time to ensure transporting the patients with trauma to appropriate departments, thereby prolonging the response time in the hospital and affecting the health outcome. In addition, the trauma centre is convenient for doctors to regularly assess and continuously improve the trauma diagnosis and treatment process, the technical operation and the clinical path; however, this setup is difficult to achieve in traditional hospital trauma treatment. These methods are positive and can be potential reasons for the successful construction of trauma centres.

However, the trauma centre in Huining County is not limited to in-hospital treatment. It also establishes a counterpart support and collaboration with primary hospitals, emphasises continuous education and establishes a regional trauma treatment system. At the same time, it establishes a mechanism and system for docking with the pre-hospital emergency system and accepting the consultation of the high-level hospitals and the treatment and referral of remote critically patients with trauma. The above measures can help solve the problem of traditional rural emergency transportation system relying excessively on hospitals at the county-level and above, thereby shortening the first-aid radius and emergency response time and improving the timeliness and health outcome of EMS-transported patients with trauma. Establishing telemedicine and internet medical channels promotes orderly visits for these patients.

Although this current study found that county-level trauma centre in China has a positive impact on the positive rate of such type of patients, constructing a county-level trauma centre is not a panacea for the regional trauma treatment system. Thus, we should be cautious about the medical centre construction policy. From the pilot of the trauma centre construction in Huining County, many advantages were observed compared with the traditional county trauma treatment system. However, the regional trauma treatment system based on trauma centre needs to be adapted to local conditions. The trauma treatment system needs to be personalised and constructed in accordance with the population size and structure of the region, medical needs, medical resource layout and government appeal. The trauma centre in Huining County is currently only established in the People’s Hospital that has the strongest medical ability in the county, mainly because the economic development and the regional medical technology of Huining County are relatively outdated in China. If many county-level trauma centres are set up, an insufficient workload and waste of medical resources will result. Conversely, if the number of trauma centres is insufficient, the needs of trauma treatment will not be met. Evidence from the United States, the United Kingdom, the Netherlands, and China [27,28,29,30] suggests that competition can improve medical quality and health outcomes. If the main motivation for high-level hospitals is to expand their patient sources, then in the long run, medical centre construction policies are unlikely to have more positive impacts on the health outcomes of patients with trauma. Excessive sorting of trauma centre can cause patients with minor trauma to be transported to higher-level trauma centres, weakening the service capacity and enthusiasm of low-level trauma centres, and insufficient sorting can cause patients with severe trauma to be transported to lower-level county-level trauma centres. China is gradually trying to categorise regional trauma centres [31]. In most cases, county-level trauma centres (level 3) and higher city-level trauma centres (levels 1 and 2) jointly provide trauma treatment. They provide continuous services to patients with trauma in their jurisdiction. The interest in competition problem may also exist within the county, and the problem of sorting patients with trauma emergency at the county hospital (level 3) and the township hospital (level 3 or level 4) will also affect the health outcome. Based on the experience of the United States, Europe, and New Zealand [32,33,34], rationally classifying patients with trauma on the basis of the graded certification standards is critical.

This study had three limitations. Firstly, before the implementation of the medical centre, the two counties did not establish a large-scale standardised hospital information system, and we were unable to collect data before 2016. Therefore, we did not examine the causal effects of medical centre policies on the health outcomes of patients with trauma transported by EMS. Secondly, data after the discharge of such patients were not obtained, thereby possibly leading to potential bias in the estimation of health outcomes. Thirdly, although we used the PSM method to control the measurable variables, other covariates may have affected the outcome variables in the samples. Considering the limitation of these variables in the case database, we did not collect the corresponding data; thus, the results may have been adversely affected. Finally, due to the availability of the data on the front page of the EHRs, this paper uses the number of disease diagnoses, inpatient care cost, admission status, and surgery to reflect the severity of the disease in the trauma inpatients, and lacks a more scientific advanced trauma scoring system to analyse the health outcomes of trauma inpatients.

## 5. Conclusions

The pilot implementation of the medical centre policy in Gansu had a positive impact on the health outcomes of patients with trauma transported by EMS. Policy makers should learn from the experience of county-level medical centre establishment in Gansu, China, and provide support for the widespread implementation of medical centres in other county-level regions.

## Figures and Tables

**Figure 1 ijerph-16-00133-f001:**
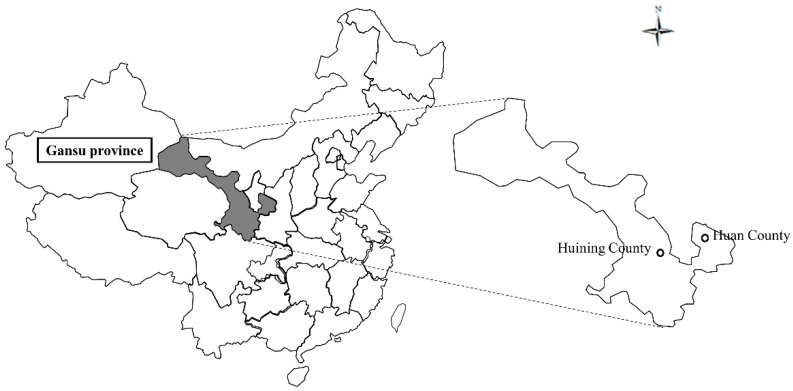
The geolocation of Huining County and Huan County in Gansu Province, China.

**Figure 2 ijerph-16-00133-f002:**
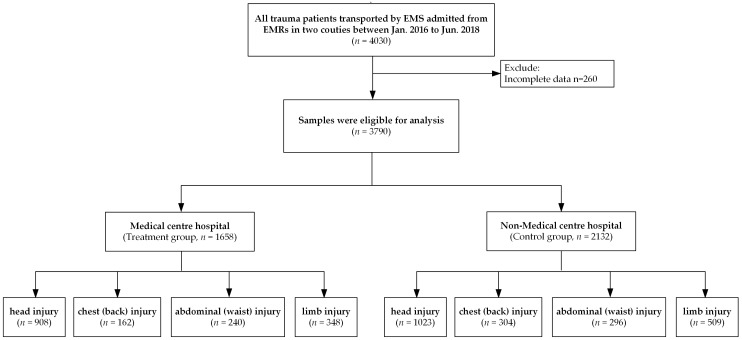
Study design and flow chart of the observations selection and the classification of those observations for propensity score matching.

**Figure 3 ijerph-16-00133-f003:**
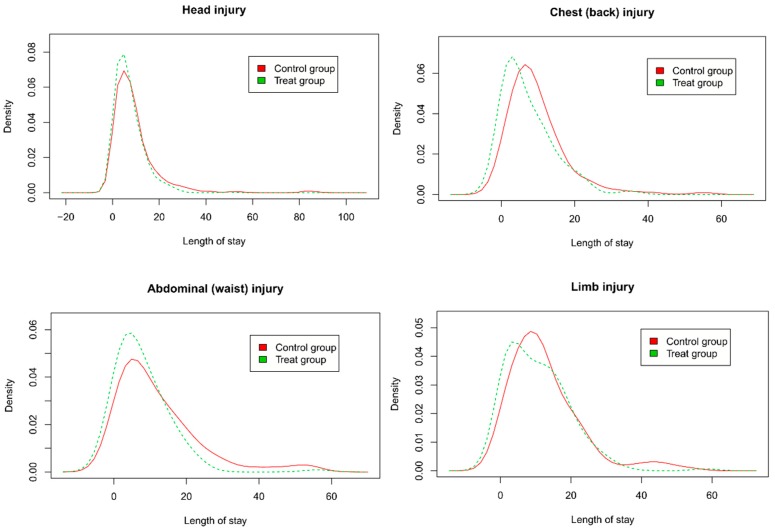
LoS of matched head injury, chest (back) injury, abdominal (waist), and limb injury patients transported by EMS.

**Figure 4 ijerph-16-00133-f004:**
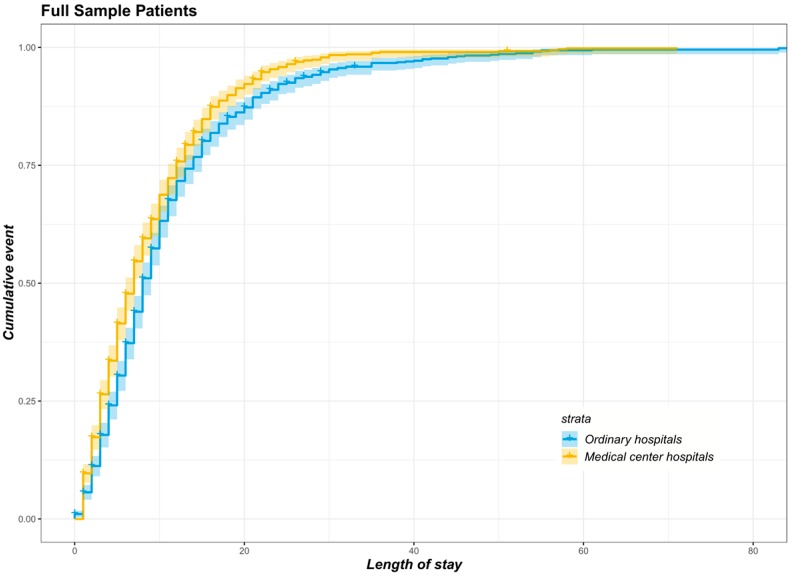
Kaplan–Meier survival curves of matched full sample patients transported by EMS in two groups.

**Figure 5 ijerph-16-00133-f005:**
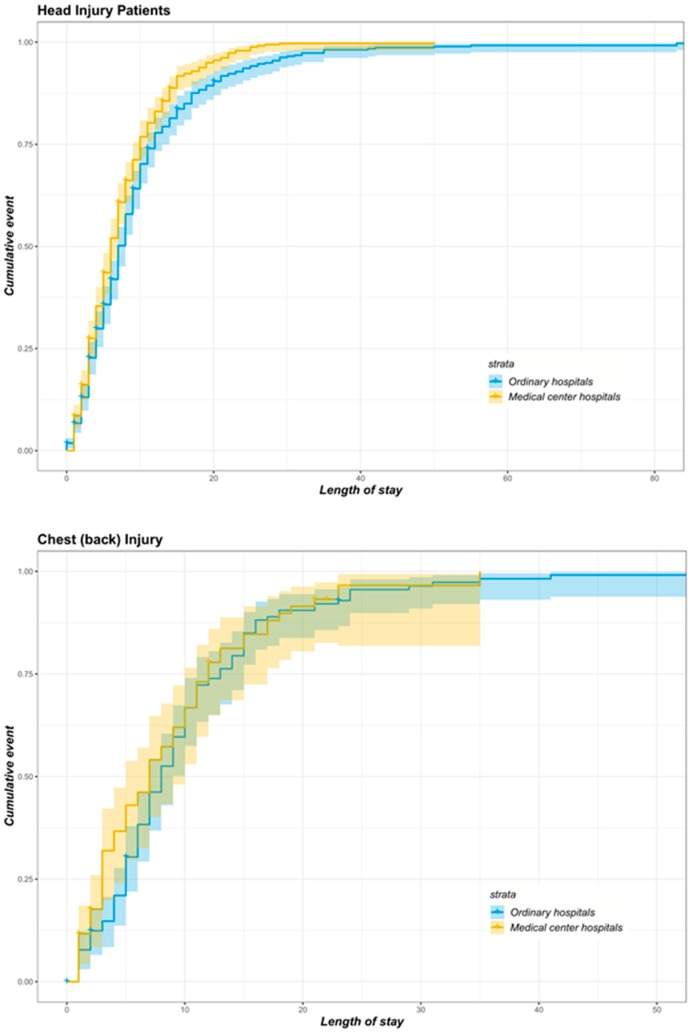
Kaplan–Meier survival curves of matched head injury, chest (back) injury, abdominal (waist), and limb injury patients transported by EMS in two groups.

**Table 1 ijerph-16-00133-t001:** Basic characteristics of the two counties and the medical capacity of two sample hospitals.

Characteristic	Huining County	Huan County
Population (thousands)	580	358
Area (square kilometres)	6439	9236
GDP (million)	61.42	74.95
Sample hospital level	Rate A, level 2	Rate A, level 2
No. of open beds per thousand people in sample hospital	1.14	1.01
No. of professional physicians per thousand people in sample hospital	1.57	1.35
No. of large medical equipment per thousand people in sample hospital	0.11	0.10

GDP, gross domestic product; No., Number. Hospital level is the evaluation index of hospital qualifications based on hospital functions, facilities, and technical strength in China.

**Table 2 ijerph-16-00133-t002:** Descriptive statistics of patient characteristics before and after propensity score matching in different groups.

Variables	Head Injury			Chest (Back) Injury			Abdominal (Waist) Injury			Limb Injury		
	Treated	Control		Treated	Control		Treated	Control		Treated	Control	
		Unmatched	Matched		Unmatched	Matched		Unmatched	Matched		Unmatched	Matched
	*n* = 908	*n* = 1023	*n* = 908	*n* = 162	*n* = 304	*n* = 162	*n* = 240	*n* = 296	*n* = 240	*n* = 348	*n* = 509	*n* = 348
Gender, *n* (%)												
Male	616 (67.8)	897 (87.7)	783 (86.2)	118 (72.8)	199 (65.4)	115 (70.9)	140 (58.3)	204 (68.9)	150 (62.7)	226 (64.9)	372 (73.0)	230 (66.2)
Female	292 (32.2)	126 (12.3)	125 (13.8)	44 (27.2)	105 (34.6)	47 (29.1)	100 (41.7)	92 (31.1)	90 (37.3)	122 (35.1)	137 (27.0)	118 (34.8)
Age group, *n* (%)												
18–40	384 (42.3)	516 (50.4)	442 (48.7)	34 (21.0)	45 (14.8)	33 (20.3)	86 (35.8)	35 (11.7)	34 (14.2)	102 (29.3)	98 (19.3)	96 (27.5)
41–50	212 (23.3)	248 (24.2)	226 (24.9)	54 (33.3)	90 (29.6)	51 (31.6)	70 (29.2)	90 (30.0)	77 (31.9)	72 (20.7)	79 (15.5)	73 (21.0)
51–60	184 (20.3)	142 (13.9)	137 (15.1)	46 (28.4)	90 (29.6)	46 (28.3)	48 (20.0)	94 (31.7)	72 (30.2)	62 (17.8)	105 (20.6)	63 (18.2)
61–70	82 (9.0)	97 (9.5)	84 (9.3)	16 (9.9)	67 (22.2)	21 (12.9)	26 (10.8)	49 (16.7)	36 (15.1)	56 (16.1)	154 (30.3)	72 (20.6)
71–80	38 (4.2)	15 (1.5)	15 (1.6)	10 (6.2)	11 (3.7)	11 (6.9)	8 (3.3)	30 (10.0)	21 (8.6)	40 (11.5)	73 (14.3)	44 (12.7)
>80	8 (0.9)	4 (0.4)	4 (0.4)	2 (1.2)	0 (0.0)	0 (0.0)	2 (0.8)	0 (0.0)	0 (0.0)	16 (4.6)	0 (0.0)	0 (0.0)
Admission status, *n* (%)												
Normal	802 (88.3)	924 (90.3)	812 (89.4)	154 (95.1)	299 (98.2)	158 (97.5)	220 (91.7)	296 (100.0)	240 (100.0)	340 (97.7)	509 (100.0)	348 (100.0)
Urgent	106 (11.7)	99 (9.7)	96 (10.6)	8 (4.9)	5 (1.8)	4 (2.5)	20 (8.3)	0 (0.0)	0 (0.0)	8 (2.3)	0 (0.0)	0 (0.0)
Inpatient care cost, mean (SD)	5507.348 (10637.5)	4568.048 (3045.821)	4908.619 (3249.856)	5423.234 (7801.27)	3604.89 (2935.47)	4303.73 (4368.813)	5270.949 (7042.654)	6705.18 (6805.462)	6678.151 (7187.803)	9294.628 (8104.878)	8857.412 (4527.157)	9347.126 (4324.453)
Surgery conducted or not, mean (SD)	0.14 (0.348)	0.22 (0.196)	0.20 (0.196)	0.15 (0.357)	0.09 (0.308)	0.11 (0.316)	0.18 (0.382)	0.26 (0.443)	0.28 (0.453)	0.59 (0.494)	0.46 (0.473)	0.49 (0.494)
Number of disease diagnoses, mean (SD)	4.53 (2.711)	3.66 (1.753)	4.26 (1.905)	4.59 (2.932)	3.65 (1.279)	4.21 (1.439)	3.92 (2.624)	3.45 (0.729)	3.42 (0.752)	4.05 (3.006)	3.56 (3.124)	3.59 (3.006)

Note: SD, standard deviation.

**Table 3 ijerph-16-00133-t003:** Maximum likelihood estimates of multivariable Cox hazard model.

Variables	All		Head Injury		Chest (Back) Injury		Abdominal (Waist) Injury		Limb Injury	
	Hazard Ratio	95% CI	Hazard Ratio	95% CI	Hazard Ratio	95% CI	Hazard Ratio	95% CI	Hazard Ratio	95% CI
Non-medical centre hospital	Ref.	Ref.	Ref.	Ref.	Ref.	Ref.	Ref.	Ref.	Ref.	Ref.
Medical centre hospital	1.249 ***	1.125–1.385	1.416 ***	1.216–1.649	1.112	0.779–1.587	1.273 **	1.021–1.848	1.078	0.838–1.387
Gender										
Male	Ref.	Ref.	Ref.	Ref.	Ref.	Ref.	Ref.	Ref.	Ref.	Ref.
Female	1.016	0.906–1.138	1.064	0.903–1.252	0.997	0.685–1.449	0.792	0.557–1.126	0.959	0.721–1.274
Age group										
18–40	Ref.	Ref.	Ref.	Ref.	Ref.	Ref.	Ref.	Ref.	Ref.	Ref.
41–50	0.971	0.845–1.116	1.164	0.962–1.408	1.004	0.611–1.652	0.910	0.579–1.429	1.337	0.911–1.962
51–60	1.037	0.893–1.205	1.224 *	0.996–1.506	1.277	0.771–2.114	1.160	0.702–1.918	0.896	0.608–1.322
61–70	0.991	0.830–1.185	1.108	0.849–1.447	1.087	0.606–1.950	2.400 ***	1.337–4.309	1.286	0.821–2.016
71–80	1.092	0.873–1.364	1.037	0.723–1.489	1.296	0.642–2.619	2.547 **	1.130–5.742	1.526 *	0.975–2.387
>80	1.086	0.724–1.630	1.688	0.887–3.213	0.993	0.285–3.455	1.198	0.161–8.933	0.697	0.346–1.401
Admission status										
Normal	Ref.	Ref.	Ref.	Ref.	Ref.	Ref.	Ref.	Ref.	Ref.	Ref.
Urgent	1.440 ***	3.069–5.799	0.755 *	0.561–1.017	1.443	0.681–3.055	0.621	0.325–1.186	1.363	0.596–3.116
Inpatient care cost	2.850 ***	2.480–3.276	4.379	3.600–5.325	5.727 ***	3.785–8.666	0.403 ***	0.312–0.522	0.544 ***	0.441–0.671
Surgery conducted or not	0.293 ***	0.214–0.400	0.953	0.684–1.328	0.527 *	0.275–1.007	2.619 ***	1.552–4.422	12.184 ***	6.783–21.885
Number of disease diagnoses	0.906 ***	0.872–0.941	0.822 ***	0.770–0.878	0.923 **	0.853–0.998	0.978	0.898–1.064	0.994	0.936–1.054
Time *admission status	0.661 ***	0.607–0.719	-	-	-	-	-	-	-	-
Time *inpatient care cost	0.886 ***	0.876–0.895	0.817 ***	0.804–0.830	0.827 ***	0.803–0.853	-	-	-	-
Time *surgery conducted or not	1.149 ***	1.122–1.175	-	-	-	-	-	-	0.869 ***	0.843–0.896
Time *number of disease diagnoses	1.008 ***	1.005–1.012	1.023 ***	1.016–1.030	-	-	-	-	-	-

Note: CI, confidence interval; “Ref.” denotes the reference group; “*” represents the product symbol of the interaction term. *** *p* < 0.01; ** *p* < 0.05; * *p* < 0.1.

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
