# Peer review of "Does County-Level Medical Centre Policy Influence the Health Outcomes of Patients with Trauma Transported by the Emergency Medical Service System? An Integrated Emergency Model in Rural China"

_ijerph, 2019, doi:10.3390/ijerph16010133_

Round 1
Reviewer 1 Report
This is a well written paper however the references are very dated & should be replaced by more contemporary literature. it is well established that minimizing time to definitive acre is critical to trauma patient outcomes, especially those requiring surgery. This paper provides an analysis of the phenomenon in rural China & reinforces studies completed elsewhere on the impact of time to definitive acre for these patients. However, there is also a developing body of literature that identifies the fact that as emergency pre-hospital services expand their interventional scope, time becomes slightly less critical for these patients providing that pre-hospital interventions all ad-value to the patients survival probability.
The paper requires a clear description of the capabilities of each of the two hospital types.
It is also not clear to me what trauma scoring system was used to include or exclude patients in this study or to stratify their outcomes - this should be corrected
The use of mortality as an outcome indicator is a very blunt measure and not appropriate where there are high rates of orthopaedic injury - these patients don't typically die in sophisticated trauma systems unless there are complication in other body systems.
Author Response
Replies to reviewer 1’s comments:
Dear Reviewer,
Thanks for your wonderful comments and suggestions on our manuscript. We have taken into account the comments and suggestions from you, in which we found most helpful. We are pleased to response to them point by point and changes in the article. We hope that the revised manuscript will satisfy you.
Comment 1:
1. This is a well written paper however the references are very dated & should be replaced by more contemporary literature.
Response: Thank you for your kind reminder. I have updated some references and marked it in red.
Comment 2:
2. The paper requires a clear description of the capabilities of each of the two hospital types.
Response: Thank you for your kind comments. Considering the availability of data, we add some basic indicators and display them in the original text in the form of table 1. Since the overall level of county hospitals in Gansu Province of China is very similar, there will be no significant differences in the ability of trauma treatment, which is also applicable in other counties of Gansu.
Comment 3:
3. It is also not clear to me what trauma scoring system was used to include or exclude patients in this study or to stratify their outcomes - this should be corrected
Response: Thank you for your kind comments. Since the data we collect are derived from EHRs, we do not include a scientific trauma scoring system. EHRs only include indicators on the number of diagnoses, LoS, inpatient care costs, and admissions status that can reflect the severity of the trauma patients transported by EMS. We use this as a basis for PSM and cox analysis. This is a limitation of this article, so I wrote this limitation in the final report of the article.
“Finally, due to the availability of the data on the front page of the EHRs, this paper uses the number of disease diagnoses, inpatient care cost, admission status and surgery to reflect the severity of the disease in the trauma inpatients, and lacks a more scientific advanced trauma scoring system to analyze the health outcomes of trauma inpatients.”
Comment 4:
4. The use of mortality as an outcome indicator is a very blunt measure and not appropriate where there are high rates of orthopaedic injury - these patients don't typically die in sophisticated trauma systems unless there are complication in other body systems.
Response: Thank you for your kind comments. Your idea is very reasonable, and we think about it when we study design. More than 90% of patients with trauma transported by EMS in China can eventually be cured or improved to meet discharge standards. We recorded an EMS-transported patient with trauma who was cured or improved to meet the discharge criteria as 1 (positive result event), whereas those who died and those who did not meet the discharge criteria were recorded as 0 (negative result event). Therefore, we are more concerned about the cure and the improvement to meet discharge standards, not death.
Reviewer 2 Report
This is a nice piece of work and well written. My only suggestion is to add some LoS data to the tables and perhaps to show a chart with LoS distributions. Average LoS can be misleading as the tail of long LoS dominates the average.
Detailed Comments:
Length of Stay (LoS) is an excellent proxy for severity and patient resilience not otherwise encapsulated in clinical codes. However, if appropriate care is delayed or not given it can also result in increased LoS.
As always average LoS can be misleading, hence, further analysis of the LoS distributions in the various groups of patients would be useful.
Indeed, if there is any before and after LoS data this would enhance the paper.
Author Response
Replies to reviewer 2’s comments:
Dear Reviewer,
Thanks for your wonderful comments and suggestions on our manuscript. We have taken into account the comments and suggestions from you, in which we found most helpful. We are pleased to response to them point by point and changes in the article. We hope that the revised manuscript will satisfy you.
Comment:
Length of Stay (LoS) is an excellent proxy for severity and patient resilience not otherwise encapsulated in clinical codes. However, if appropriate care is delayed or not given it can also result in increased LoS. As always average LoS can be misleading, hence, further analysis of the LoS distributions in the various groups of patients would be useful. Indeed, if there is any before and after LoS data this would enhance the paper.
Response: Thank you for your kind comment. What you said makes sense. Inadequate construction of the medical center may lead to inconsistency in the treatment process, resulting in delayed or no treatment of trauma treatment transported by EMS, resulting in an increase in LoS. Therefore, this paper uses kernel density plot to explain the distribution of LoS of four types of trauma patients transported by EMS in section 3.2, which is of great help to this paper. In the method part, the significance and method of analyzing the distribution of LoS are briefly introduced.
Reviewer 3 Report
The article is hard to follow and understand, with the methods being unclear, mainly on the control and matching aspects, which leads to a possible bias in the results.
Basically the paper is ok, but the statistics are not conclusive. I don’t understand the methodology. So they can add a proper methodology section supervised by a statistician.
Author Response
Replies to reviewer 3’s comments:
Dear Reviewer,
Thanks for your wonderful comments and suggestions on our manuscript. We have taken into account the comments and suggestions from you, in which we found most helpful. We are pleased to response to them point by point and changes in the article. We hope that the revised manuscript will satisfy you.
Comment:
The article is hard to follow and understand, with the methods being unclear, mainly on the control and matching aspects, which leads to a possible bias in the results. Basically the paper is ok, but the statistics are not conclusive. I don’t understand the methodology. So they can add a proper methodology section supervised by a statistician.
Response: Thank you for your kind comments. This is indeed a flaw in this paper. The description of the PSM method is too general and may make the reader not quite understand the process of the analysis method. Therefore, this paper further explains the methods of matching and control in the analysis method introduction of the paper, including how to ensure and test the balance between the covariates of the two sets of data and the overall model. We have dealt with this process to minimize the bias of health outcome in two groups, to ensure the reliability of the results.
Round 2
Reviewer 3 Report
Thank you for addressing our comments